

# Physiological thermal responses of three Mexican snakes with distinct lifestyles

Ricardo Figueroa-Huitrón[1,2], Anibal Díaz de la Vega-Pérez[3],
Melissa Plasman[4] and Hibraim Adán Pérez-Mendoza[1]

[1] Laboratorio de Ecología Evolutiva y Conservación de Anfibios y Reptiles, Facultad de Estudios Superiores Iztacala, UNAM, Tlalnepantla, Estado de México, Mexico
[2] Posgrado en Ciencias Biológicas, Unidad de Posgrado, Universidad Nacional Autónoma de México, Ciudad de México, Mexico
[3] Consejo Nacional de Humanidades Ciencias y Tecnologías-Centro Tlaxcala de Biología de la Conducta, Universidad Autónoma de Tlaxcala, Tlaxcala, Mexico
[4] Centro Tlaxcala de Biología de la Conducta, Universidad Autónoma de Tlaxcala, Tlaxcala, Mexico

Corresponding author
Hibraim Adán Pérez-Mendoza,
hibraimperez@ciencias.unam.mx

## ABSTRACT

The impact of temperature on reptile physiology has been examined through two main parameters: locomotor performance and metabolic rates. Among reptiles, different species may respond to environmental temperatures in distinct ways, depending on their thermal sensitivity. Such variation can be linked to the ecological lifestyle of the species and needs to be taken into consideration when assessing the thermal influence on physiology. This is particularly relevant for snakes, which are a very functionally diverse group. In this study, our aim was to analyze the thermal sensitivity of locomotor performance and resting metabolic rate (RMR) in three snake species from central Mexico (*Crotalus polystictus*, *Conopsis lineata*, and *Thamnophis melanogaster*), highlighting how it is influenced by their distinctive behavioral and ecological traits. We tested both physiological parameters in five thermal treatments: 15 °C, 25 °C, 30 °C, 33 °C, and 36 °C. Using the performance data, we developed thermal performance curves (TPCs) for each species and analyzed the RMR data using generalized linear mixed models. The optimal temperature for locomotion of *C. polystictus* falls near its critical thermal maximum, suggesting that it can maintain performance at high temperatures but with a narrow thermal safety margin. *T. melanogaster* exhibited the fastest swimming speeds and the highest mass-adjusted RMR. This aligns with our expectations since it is an active forager, a high energy demand mode. The three species have a wide performance breadth, which suggests that they are thermal generalists that can maintain performance over a wide interval of temperatures. This can be beneficial to *C. lineata* in its cold habitat, since such a characteristic has been found to allow some species to maintain adequate performance levels in suboptimal temperatures. RMR increased along with temperature, but the proportional surge was not uniform since thermal sensitivity measured through $Q_{10}$ increased at the low and high thermal treatments. High $Q_{10}$ at low temperatures could be an adaptation to maintain favorable performance in suboptimal temperatures, whereas high $Q_{10}$ at high temperatures could facilitate physiological responses to heat stress. Overall, our results show different physiological adaptations of the three species to the environments they inhabit. Their different activity patterns and foraging habits are closely linked to these adaptations. Further studies of other populations with different climatic conditions would provide

valuable information to complement our current understanding of the effect of environmental properties on snake physiology.

# INTRODUCTION

Temperature affects the biology of ectotherms at all levels, from physiology to behavior, and these animals have developed various adaptations to balance such effects (*Abram et al., 2017*). In this sense, different species may respond to environmental temperatures in distinct ways, depending on their sensitivity to environmental temperatures. The extent to which temperature influences the physiological performance of an organism is called thermal sensitivity, a continuum ranging from thermal specialists whose performance is strongly influenced by temperature, to thermal generalists that perform well over a broad range of temperatures (*Angilletta, 2009*). How sensitive a species is to temperature can be linked to behavioral, ecological, and evolutionary aspects (*Bars-Closel, Camacho & Kohlsdorf, 2018*; *Senior et al., 2019*). In this context, it is of particular interest to determine the magnitude and consequences of the effects of temperature on physiology, and this has been one of the focal points of research in non-avian reptiles (hereinafter "reptiles").

The impact of temperature on physiology has been examined through two main parameters: performance, primarily assessed through locomotion speed, and metabolic rates, primarily evaluated by the rates of $O_2$ consumption or $CO_2$ production (see *Bennett, 1990*; *Angilletta, Niewiarowski & Navas, 2002*; *de Andrade, 2016*; *Tomlinson, 2019*). The thermal sensitivity of performance is analyzed by developing thermal performance curves (TPCs), which are continuous equations that describe how performance changes as a function of body temperature (*Taylor et al., 2021*). TPCs have a distinctive shape, and they follow a pattern in which performance increases along temperature up to an optimal level beyond which it decreases rapidly (*Angilletta, 2006*; *Tomlinson, 2019*). Useful parameters and information can be obtained after generating a TPC, such as the optimal temperature at which performance is maximized ($T_o$) and performance breadth ($B_{80}$), which is the temperature interval in which favorable performance (usually specified as 80% of maximum performance) can be maintained (*Angilletta, Niewiarowski & Navas, 2002*). Furthermore, the development of TPCs also encompasses the determination of the thermal tolerance breadth (TTB) of the organisms, which is delimited by the critical maximum and minimum temperatures ($CT_{max}$ and $CT_{min}$). These values represent the threshold beyond which the physiological functioning of an organism is lost and mark the start and end points of the TPCs, where performance drops to zero (*Taylor et al., 2021*). $CT_{max}$ is also the reference point for determining the thermal safety margin (TSM), a concept defined by *Sinclair et al. (2016)* as the difference between $T_o$ and $CT_{max}$. The TSM represents thus the margin available to organisms to sustain their performance prior to reaching their thermal threshold. In this way, TPCs provide valuable insights into the relationship between temperature and physiological performance.

Metabolic rate is also strongly influenced by temperature, and resting metabolic rate (RMR) is one of the most commonly used measurements to assess this relation, as it increases with body temperatures. Different environmental conditions can affect metabolic rates distinctively, leading to variable energetic costs at different temperatures. For instance, species living in arid environments present low metabolic rates, a strategy that may decrease energetic costs and dehydration risk (*Dupoué, Brischoux & Lourdais, 2017*). On the other hand, species that live in cold habitats often have a higher RMR than those from temperate or warm environments (*Žagar et al., 2018*; *Plasman et al., 2020*), an adaptation that enables them to maintain high activity levels in non-favorable environments, where the conditions can impair locomotor performance (*Stellatelli et al., 2022*). The thermal sensitivity of metabolic rates has often been represented through the $Q_{10}$ index. This index represents how much metabolic rates change with a 10 °C increase in body temperature. A $Q_{10}$ value of two, for example, represents that the metabolic rate has doubled (*Schmidt-Nielsen, 1997*). In reptiles, the most common scenario is that species present a $Q_{10}$ within the range of 2–3 (*Kinoshita et al., 2018*). However, a wide range of variation has been reported in the group (*Frappell & Daniels, 1991*; *Brashears & Denardo, 2013*; *Arnall, Kuchling & Mitchell, 2014*).

The physiology of reptiles and their relationship with environmental temperatures are closely linked to the habits of a species. In a comparative study among squamates, *Andrews & Pough (1985)* report that fossorial species are the ecological group with the lowest metabolic rates, although some species do not follow this trend (*Wang & Abe, 1994*). As for aquatic reptiles, there is no clear evidence of differences between their metabolic rates and those of terrestrial reptiles (*Christian et al., 1996*; *Wallace & Jones, 2008*). However, being highly thermally sensitive can be beneficial to aquatic species, like the yellow-lipped Sea Kraits (*Laticauda colubrina*), whose metabolic rate increases on land, boosting processes like digestion, and decreases in water, allowing them to spend less energy while submerging and foraging (*Dabruzzi, Sutton & Bennett, 2012*). In the terrestrial snake *Storeria dekayi*, performance level increases with higher temperatures, but the degree of thermal sensitivity varies across different microhabitats and locomotion modes (*Gerald & Claussen, 2007*). Furthermore, there are also metabolic differences regarding foraging mode. Active foragers require more energy to pursue prey, which leads them to have higher metabolic rates than ambush predators (*Aubret, Tort & Sarraude, 2015*; *Dupoué, Brischoux & Lourdais, 2017*).

In this context, it is important to take into consideration the variation in ecological and behavioral traits when assessing the effects of temperature on reptile physiology. This is of particular relevance when working with snakes, which are a very diverse group, both in species richness and functional diversity (*Tingle, Garner & Astley, 2024*). Despite the great diversity present in Mexico (*Flores-Villela & García-Vázquez, 2014*; *Ramírez-Bautista et al., 2023*), to our knowledge, there has been no study focusing on the thermal sensitivity of physiological traits in Mexican snakes. Here, our aim was to analyze the thermal sensitivity of locomotor performance and RMR in three snake species from central Mexico (*Crotalus polystictus*, *Conopsis lineata*, and *Thamnophis melanogaster*), highlighting how it is influenced by their distinct behavioral and ecological traits. We selected these species as

they are distinctive representatives of the herpetofaunal communities in the region and are among the most abundant species. Moreover, they encompass three prevalent lifestyles commonly found within snake assemblages (*Di Pietro et al., 2020*): *C. lineata* is a small fossorial ambush predator predominantly dwelling underground; *C. polystictus* is a terrestrial ambush predator commonly found in grasslands and beneath rocks; and *T. melanogaster* is a semiaquatic active forager, typically inhabiting lakeshores and streams. We hypothesized that, since they have distinct lifestyles, the physiological thermal responses would be different. We predicted that, considering it is an active forager that spends considerable time in the open, *T. melanogaster* would be a thermal specialist and present the highest levels of performance (higher $T_o$ and $B_{80}$) and RMR (after adjusting for body mass). On the contrary, being a fossorial species with little mobility, we predicted *C. lineata* to present the lowest levels of performance and RMR. Since it is an ambush predator, albeit with more conspicuous habits than *C. lineata*, we expected *C. polystictus* to have intermediate levels of performance and RMR relative to the other two species.

## MATERIALS AND METHODS

### Study species and sites

*Crotalus polystictus* is an endemic rattlesnake from central Mexico (*Mociño-Deloya et al., 2009*). It is a diurnal, medium-sized viper with an average snout-vent length (SVL) of about 60 to 70 cm (*Campbell & Lamar, 2004*). It inhabits pine-oak forests, scrublands, and both dry and humid grasslands between 1,450 and 2,739 m above sea level (*Meik et al., 2012*). This terrestrial species spends considerable time under rocks, and it alternates between shelters and the open for thermoregulation. It feeds mostly on small rodents as a sit-and-wait predator (*Mackessy et al., 2018*). The study site is located in San Bartolo Morelos, on the northwest side of Estado de México. The site is composed of a series of croplands surrounded by grasslands and oak forest patches, and it is located at an elevation of 2,660 m. Climate is subhumid temperate with summer rains. Rocky areas between croplands provide suitable shelters for the rattlesnakes.

*Conopsis lineata* is a small-sized fossorial snake with a mean SVL of around 18 cm. This diurnal species spends most of its time within burrows or under rocks and logs, where it feeds on larvae, insects, and other arthropods (*Ramírez-Bautista et al., 2009*; *García-Balderas et al., 2014*). It inhabits pine-oak forests, montane cloud forests, xerophytic shrubs, and fir forests in an altitude range from 1,700 to 3,100 m above sea level (*Goyenechea & Flores-Villela, 2006*; *Ramírez-Bautista et al., 2009*). The study site is located in Los Dinamos National Park, which is part of the forested and montane area in the western part of Mexico City. Climate is subhumid semicold with annual rainfall of 900 to 1,300 mm and a mean annual temperature range of 9 °C to 15 °C (*Kovács et al., 2021*). The study site is an open area used for shepherding and plum tree cultivation, located at an elevation of 2,720 m.

*Thamnophis melanogaster* is a medium-sized, semiaquatic colubrid with an average SVL of about 40 cm (*Manjarrez, Macías Garcia & Drummond, 2017*). It is distributed widely throughout Mexico, and it inhabits riverbanks and lake shores (*Ramírez-Bautista et al., 2009*). It uses shelters on land, near aquatic bodies, and forages actively underwater,
feeding on aquatic prey such as tadpoles, fish, and invertebrates (*Manjarrez, García & Drummond, 2013*). It is mostly a diurnal species but can also be active during particularly warm nights (*Rossman, Ford & Seigel, 1996*). The study site is the ecotourism park Arcos del Sitio, which is located within the Tepotzotlán mountain range in Estado de México, at an elevation of 2,280 m. Climate is subhumid temperate with summer rains, with an annual precipitation of 700–800 mm (*Espinosa-Graciano & García-Collazo, 2017*). The dominant vegetation in the area is crasicaule shrubland, heavily fragmented by shepherding areas. Gallery forest is present along the Arcos del Sitio River, where individuals of *T. melanogaster* are mostly found in areas with reduced vegetation cover.

## Field work and snake captivity

We visited each site every 2 months from April 2021 to July 2023, excluding the winter period (December to February) when snakes are rarely found active. Furthermore, many squamate species present a decrease in metabolic rates during winter as part of the brumation process (*Dubiner et al., 2023*), which would have biased our measurements. For *C. polystictus* and *T. melanogaster*, each visit lasted 3 days, during which a team of 2–4 people searched for the snakes in their preferred microhabitats from 10:00 to 18:00 h. As the study site of *C. lineata* is considerably smaller, sampling occasions for this species were shorter, lasting 1 or 2 days, with a sampling schedule from 10:00 to 14:00 h. For each snake encountered, we registered the time of capture, body mass (g), SVL (mm), sex, and geographic coordinates. We determined sex by cloacal probing; however, to avoid hurting them, we did not use the technique on the small juveniles, whose size prevented the probe from entering easily. We identified pregnant females by palpation, which were not considered for the experiments. After capturing the snakes, we kept them inside cloth sacks and transported them to the laboratory. We planned our field trips and experiments so that all individuals spent a similar amount of time in captivity (between 5 and 8 days) before the onset of the experiments, which lasted between 7 and 10 days. On average, the time between capture and the end of the experiments was just over 2 weeks.

While in captivity, we maintained the snakes in either 40 × 30 × 30 cm terrariums (*C. polystictus* and *C. lineata*) or in 50 × 40 × 30 cm fish tanks conditioned with a dry area (*T. melanogaster*). We placed the enclosures in a room at ambient temperature with natural light, under a natural photoperiod. We conditioned all enclosures with cardboard boxes to serve as refuges and water bowls that we filled constantly. We placed heating mats under one extreme of each enclosure so that snakes could have the opportunity to thermoregulate. Using a digital timer, we set the heating mats to be on at a constant schedule of 09:00 to 17:00 h. We did not feed the snakes prior to the experiments so that digestion did not interfere with the tests, which is a safe procedure since snakes can spend extended periods of time without eating (*McCue, Lillywhite & Beaupre, 2012*). After the experiments were finished, we maintained the snakes in their enclosures and fed them once every 2 weeks until the subsequent field trip, during which we liberated the organisms at their respective study site. We fed *C. lineata* with crickets (*Acheta domesticus*), *C. polystictus* with mice (*Mus musculus*), and *T. melanogaster* with small fish (*Chirostoma* sp.).

Permission to collect the animals was given by the Secretary of Environment and Natural Resources (SEMARNAT) through the permits SGPA/DGVS/06935/21 and SPARN/DGVS/00316/23. Experimental procedures were made under the approval of the ethics committee of the Faculty of Higher Studies (FES) Iztacala of the National Autonomous University of Mexico (UNAM) through the permit CE/FESI/022023/1578.

## Thermal tolerance

To avoid subjecting a large number of animals to stressful experimentation, we decided to perform the $CT_{min}$ and $CT_{max}$ trials with a subset of individuals. We utilized the snakes collected in 2023: 10 individuals of *C. polystictus* (six females and four males), eight of *C. lineata* (three females, two males, and three juveniles), and seven of *T. melanogaster* (three females and four males). We followed the most commonly used procedure to determine thermal tolerance in reptiles, which is to heat and cool the animals to the point of losing the righting response (*Taylor et al., 2021*). To avoid confounding negative effects, we made these experiments last after performance and RMR tests. To determine $CT_{min}$, we placed the snakes in a $50 \times 40 \times 50$ cm plastic container, of which the bottom was covered with frozen refrigerant gel packs. With this procedure, snakes were cooled at a rate of $1\,°C/min$. After 15 min, using a pair of 30 cm metal tweezers, we started turning snakes on their backs every minute. At the moment in which the snakes were unable to right themselves, we terminated the experiment and recorded their body temperature with a contact thermometer (Fluke model 52-II, $\pm 0.1\,°C$), whose sensor was inserted 0.5–1 cm into the cloaca, and recorded it as the $CT_{min}$. To measure $CT_{max}$, we placed a 200 W heat lamp 50 cm above a plastic bucket (30 cm in diameter and 40 cm high), in which we placed the snakes. With these conditions, we were able to heat the snakes steadily at a rate of $1.5\,°C/min$. When the snakes recurred to panting as a cooling strategy, we started turning them on their backs every 30 s. When they were unable to right themselves up, we recorded that temperature as $CT_{max}$, in the same way as we did with $CT_{min}$. Right after each $CT_{min}$ or $CT_{max}$ test ended, we placed the snakes in direct contact with water at ambient temperature (1 cm deep) inside a $60 \times 40 \times 50$ cm plastic container and allowed them to recover. To avoid subjecting the snakes to consecutive stressful experiments, we first performed the $CT_{min}$ tests and let the snakes recover for 24 h before the $CT_{max}$ tests. With these data, we determined TTB as the distance in $°C$ between $CT_{max}$ and $CT_{min}$ (*Stellatelli et al., 2022*; *Valdez Ovallez et al., 2022*).

## Thermal treatments preparation

We ran both the performance and RMR trials on a set of thermal treatments that, without exceeding the TTB limits, encompasses temperatures below, within, and over their set point temperature range ($T_{set}$). This range represents the target body temperatures that organisms attempt to achieve (*Hertz, Huey & Stevenson, 1993*), and we calculated it for the three species in another study focused on their thermal ecology (R. Figueroa-Huitron et al., 2023, unpublished data). We found that, while their overall thermoregulation strategies are different, their $T_{set}$ and field body temperatures are similar. This allowed us to make a

generalization to determine five ecologically relevant temperatures for the three species: 15 °C, 25 °C, 30 °C, 33 °C, and 36 °C. Approximately, 15 °C is the lowest body temperature we registered on the field, 25 °C is near the mean $T_{set}$, 30 °C is the upper limit of $T_{set}$, and 33 °C is the maximum body temperature registered in the field. Though it was not considered at first, we included the 36 °C treatment subsequently to observe the response of the snakes above voluntarily maintained body temperatures in natural conditions and to better adjust the thermal performance curves over a wide temperature range (*Taylor et al., 2021*).

Before the onset of each test, we set the snakes to acclimatize to the target temperature for at least 30 min. To achieve this, we put the individuals in hermetically closed chambers placed in a darkened incubator. Depending on the size of each individual, we used 50 ml centrifuge tubes or plastic containers of 300 or 900 ml, which allowed the snakes to adopt a coiled resting position but not move freely. In both experimental procedures, we randomized the sequence of thermal treatments for each individual so that none passed through the treatments in direct ascending or descending order. Since metabolic rates change during rest hours (*Dubiner et al., 2023*), all tests were performed between 9:00 and 18:00 h, within the observed activity period of the snakes in the field, so that our data accurately represent RMR. Individuals performed only one experimental trial per day to reduce possible confounding effects derived from acute stress.

## Locomotor performance

We evaluated locomotor performance through swimming speed. We decided to use swimming speed over crawling speed because in terrestrial conditions snakes frequently adopt defensive behaviors, such as coiling and biting, which would interfere with the locomotion trials. Swimming forward is a reflex behavior of snakes in water; thereby, swimming speed is an adequate measurement to assess the locomotor capacity of both semiaquatic and terrestrial snakes (*Aïdam, Michel & Bonnet, 2013*). To carry out the swimming speed trials, we designed a rectangular lane made from wood, covered with impermeable resin. The lane was 350 cm long, 30 cm wide, and 40 cm high. Along the lane, we placed marks every 30 cm to serve as a reference for the determination of the distance covered by the snakes. For each thermal treatment, we filled the lane with water up to a level of 15 cm and either heated or cooled the water up to the targeted trial temperature using refrigerated gel packs and electric water heaters.

On each trial, we placed the snake at one end of the lane and followed it as it went down the lane. If a snake stopped, we stimulated it to continue swimming forward by gently touching it with a herpetological hook. We made sure that snakes advanced at least through five marks (150 cm) throughout the trials, which lasted a maximum of 2 min. We filmed trials from an overhead view with a digital video camera (Akaso EK7000) at the highest frame rate: 60 frames per second (fps). We analyzed the resulting videos in the video editing program Shotcut (Version 21.05.18), in which we determined the number of frames it took for the snakes to move through each 30 cm segment. We then divided each value by 60 to get time in seconds, and with that information, we calculated swimming

speed for each segment. We chose the highest value as the maximum swimming speed at the given temperature (Table S1).

## Resting metabolic rate

To measure RMR, we used the rate of $CO_2$, since it is a reliable measure of basic maintenance in a steady state (*Holden et al., 2022*). We used an open flow respirometry system, following the protocol of *Plasman et al. (2020)*. Incurrent air was scrubbed with Drierite (calcium sulfate for water vapor) and Ascarite (sodium hydroxide for $CO_2$) so that dry, $CO_2$-free air was pumped into the chambers at a rate of 100 ml·min$^{-1}$. Water vapor was also removed from the excurrent air sample, and the proportion of $CO_2$ in the excurrent gas was analyzed using a Foxbox $O_2/CO_2$ analyzer (Sable Systems International). After the acclimation period, the trials ran over 30 min, unless more time was required for the $CO_2$ levels to stabilize. Due to logistical difficulties, we did not perform trials for the 33 °C treatment with the animals collected in 2023 (10 *C. polystictus*, eight *C. lineata*, and seven *T. melanogaster*). We utilized the Expedata software (Sable Systems International, North Las Vegas, NV, USA) to visualize the $CO_2$ curves and to make the calculation of the lowest average levels of $CO_2$ during 100 continuous seconds. From the data obtained, we calculated RMR in terms of rates of carbon dioxide production ($VCO_2$; ml $CO_2$/min) using the following equation (*Lighton, 2008*),

$$VCO_2 = FR_i(F_eCO_2 - F_iCO_2)$$

where $FR_i$ is the incurrent air flow rate in ml/min, and $F_eCO_2$ and $F_iCO_2$ are the fractional concentrations of $CO_2$ in the excurrent and incurrent gas samples, respectively. Finally, we calculated the thermal sensitivity through the $Q_{10}$ index. We calculated $Q_{10}$ both over the entire thermal interval of the experiments and between subsequent thermal treatments using the following equation:

$$Q_{10} = (RMR_2/RMR_1)^{[10/(T_2 - T_1)]}$$

## Statistical analyses

To assess the thermal sensitivity of locomotor performance, we constructed TPCs. For each species, we utilized the average swimming speed of all individuals at each of the five thermal trials as the base data on which the curves were fitted. The boundaries of the curves, where the performance value is zero, were set at the mean $CT_{min}$ and $CT_{max}$. Following common practice in the field (see *Angilletta, 2006*; *Taylor et al., 2021*), we fitted a set of curve equations and obtained their $R^2$ values and the Akaike information criterion adjusted for small samples (AICc, our sample size was 33 for *C. polystictus*, 31 for *C. lineata* and 38 for *T. melanogaster*). To fit the curves, we utilized the software TableCurve 2D (Systat Software Inc., version 5.01). From the pool of fitted curves, we filtered the ones with the highest $R^2$ (over 0.9) and selected the best curve according to the lowest AICc (Table S2) as their TPC. We then calculated maximum swimming speed ($V_{max}$), $T_o$, TSMs, and $B_{80}$ from the TPC of each species.

We analyzed the data on resting metabolic rates using generalized linear mixed models (GLMMs), which allow for study designs with repeated measures. We $\log_{10}$ transformed RMR and body mass to normalize the data and account for the non-linearity of their relationship (*Glazier, 2021*). We set thermal treatment, body mass, and sex as fixed predictors and snake identification as the random factor. We tested seven models with combinations of these predictors, including the interaction of temperature and body mass (Table S3). The best model was selected based on the lowest AICc.

To account for body mass while testing for differences in RMR between species, we ran a GLM with species set as a fixed factor and tested the effect of the species × body mass interaction. We then calculated the estimated marginal means and performed marginal contrasts analysis. To test for differences in swimming speeds, direct RMR among species, and overall $Q_{10}$, we used the Kruskal-Wallis and Dunn tests. We tested normality with the Shapiro-Wilk test and homoscedasticity with the Bartlett test. Analyses were made in R version 4.3.2 (*R Core Team, 2023*) using the following packages: AICcmodavg (*Mazerolle, 2023*) for computing AICc values; FSA (*Ogle et al., 2023*) for performing the Kruskal-Wallis and Dunn tests; lme4 (*Bates et al., 2015*) for fitting GLMMs; and modelbased (*Makowski et al., 2020*) for estimating and comparing marginal means. We used ggplot2 (*Wickham, 2016*) to generate the figures.

# RESULTS

## Locomotor performance and thermal tolerance

We examined the effect of temperature on performance on 33 *C. polystictus* (12 males, 14 females, and seven unsexed juveniles), 31 *C. lineata* (16 males, 15 females), and 38 *T. melanogaster* (12 males, 15 females, and 11 juveniles). The curves that best fit the performance data were an exponentially modified Gaussian for *C. polystictus* ($R^2$ = 0.92, Table S2), an asymmetric logistic for *C. lineata* ($R^2$ = 0.94, Table S2), and an extreme value distribution for *T. melanogaster* ($R^2$ = 0.93, Table S2). The TPCs of the three species are represented in Fig. 1. Overall, *T. melanogaster* reached the highest swimming speeds, whereas *C. lineata* presented the lowest ($X^2$ = 176.02, $P$ < 0.001). *C. polystictus* presents the highest $T_o$, followed by *T. melanogaster* and then *C. lineata* (Table 1). The TTB of *C. polystictus* is 37.2 °C, ranging from 5.9 °C to 43.1 °C (Table 1). The $T_o$ (36.7 °C) and the upper limit of the $B_{80}$ (39.2 °C) of this species are close to the $CT_{max}$, which resulted in the lowest TSM among the three species (Fig. 1, Table 1). The TTB of *C. lineata* is 31.9 °C, ranging from 9.3 to 41.2 °C, whereas that of *T. melanogaster* is 33.1 °C, ranging from 8.5 °C to 41.6 °C (Table 1). These two species present similar values of $T_o$ (27.8 °C, 28.6 °C) and $B_{80}$ (20.1–34.3 °C, 20.7–34.9 °C, Table 1).

## Resting metabolic rate

We calculated the RMR of 41 *C. polystictus* (15 males, 19 females, and seven unsexed juveniles), 26 *C. lineata* (11 males, 13 females, and two juveniles), and 36 *T. melanogaster* (13 males, 15 females, and eight juveniles). The bigger species had higher absolute values of RMR, with *C. polystictus* presenting the highest values of RMR ($X^2$ = 149.18, $P$ < 0.001; Figs. 2, 3; Table S4). However, after variable transformation and taking body mass into

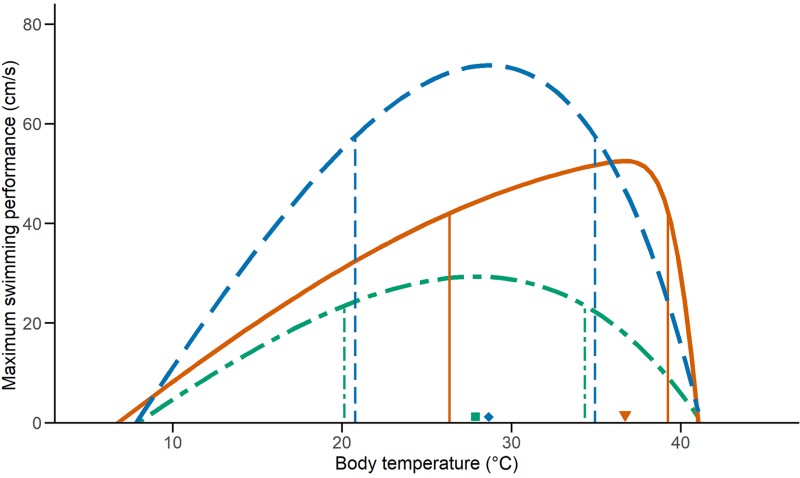

**Figure 1 Thermal performance curves of the three study species.** Vertical lines indicate the inferior and superior limits of $B_{80}$, and the shapes point to $T_o$. Solid lines and a triangle, in orange, represent *Crotalus polystictus*. Dot-dashed lines and a square, in green, represent *Conopsis lineata*. Dashed lines and a diamond, in blue, represent *Thamnophis melanogaster*.

**Table 1 Performance characteristics.**

|  | $CT_{min}$ (°C) | $CT_{max}$ (°C) | TTB (°C) | $T_o$ (°C) | TSM (°C) | $V_{max}$ (cm/s) | $B_{80}$ (°C) |
|---|---|---|---|---|---|---|---|
| *Crotalus polystictus* | 5.9 ± 0.27 | 43.1 ± 0.92 | 37.2 ± 0.83 | 36.7 | 6.4 | 42.0 | 26.3–39.2 |
| *Conopsis lineata* | 9.3 ± 1.67 | 41.2 ± 1.42 | 31.9 ± 2.19 | 27.8 | 13.4 | 29.3 | 20.1–34.3 |
| *Thamnophis melanogaster* | 8.5 ± 1.14 | 41.6 ± 1.34 | 33.1 ± 1.60 | 28.6 | 13 | 71.7 | 20.7–34.9 |

**Note:**
Relevant performance characteristics of the study species. $CT_{min}$ and $CT_{max}$ values presented ± SD.

consideration, *T. melanogaster* had a higher estimated marginal mean (t = −2.56, $P < 0.05$). No significant differences were found between *C. polystictus* and *C. lineata*.

For all three species, the GLMM that best explained RMR included the additive effect of temperature and body mass (RMR~Temperature + Body mass. Conditional $R^2$ = 0.96 for *C. polystictus*, 0.85 for *C. lineata*, and 0.92 for *T. melanogaster*). The coefficients of the best models for the three species are presented in Table 2, whereas all tested models with their AICc values are presented in Table S3. In all three species, the thermal treatments affected RMR positively, and this effect increased along with temperature (Table 2). Body mass also had a positive, significant effect on RMR in the three species.

*Conopsis lineata* and *T. melanogaster* had very similar values of overall $Q_{10}$ (2.20, 2.11, Table 3), but *C. polystictus* had a significantly lower $Q_{10}$ than the other two species (1.62, $X^2$ = 28.447, $P < 0.001$; *post-hoc* Dunn test: $P < 0.001$ in both cases). The variation of $Q_{10}$ across the thermal treatments was also similar in *C. lineata* and *T. melanogaster*. In both species, $Q_{10}$ progressively decreased from 15 °C to 33 °C but increased abruptly in the 33 °C to 36 °C interval (Table 3). On the other hand, $Q_{10}$ of *C. polystictus* increased slightly from 15 °C to 33 °C but also experienced a noticeable increment between 33 °C and 36 °C (Table 3).

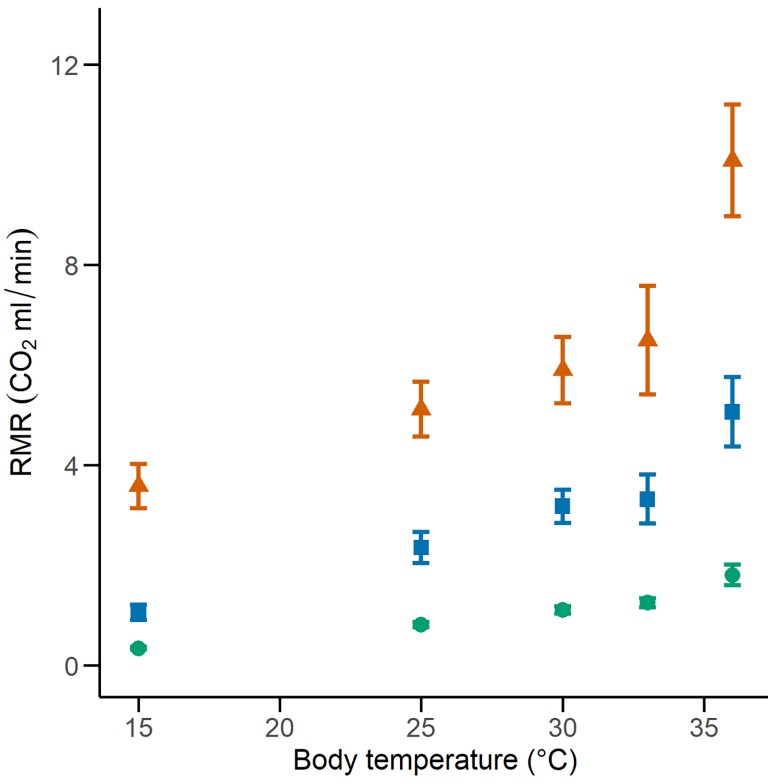

**Figure 2 Resting metabolic rates for the studied species.** RMR per thermal treatment of the three study species. Orange triangles represent *Crotalus polystictus*, green circles represent *Conopsis lineata*, and blue squares represent *Thamnophis melanogaster*. Horizontal lines above and below each point represent ±1 SE.

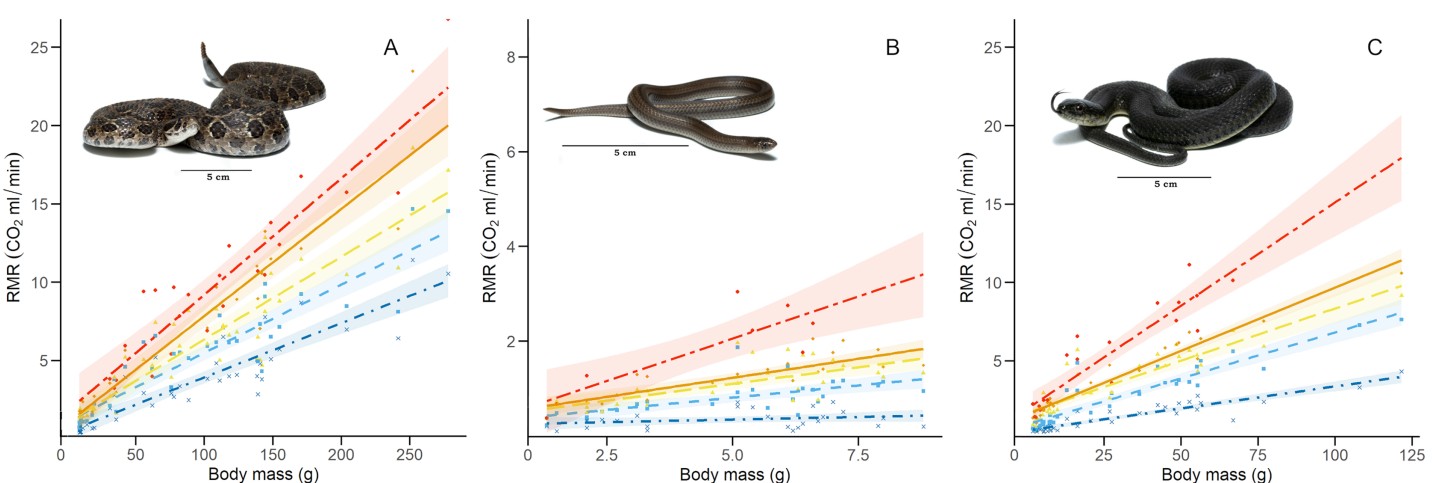

**Figure 3 Relation between resting metabolic rate and body mass.** Relation of RMR with body mass at each thermal treatment for the three study species: (A) *Crotalus polystictus*, (B) *Conopsis lineata*, and (C) *Thamnophis melanogaster*. Visual representation of the thermal treatments is as follows (from bottom to top): 15 °C: dot-dashed line and crosses in dark blue. 25 °C: dashed line and squares in light blue. 30 °C: long-dashed line and triangles in yellow. 33 °C: solid line and diamonds in orange. 36 °C: unevenly dashed line with circles in red. Shaded areas around each line represent 95% CI. Photographs credits: L. E. Bucio-Jiménez.

**Table 2  Resting metabolic rate models.**

| | *Crotalus polystictus* | | | | *Conopsis lineata* | | | | *Thamnophis melanogaster* | | | |
|---|---|---|---|---|---|---|---|---|---|---|---|---|
| | RMR~Temperature + Body mass | | | | RMR ~ Temperature + Body Mass | | | | RMR~Temperature + Body mass | | | |
| Fixed effects | Estimate ± SE | df | t value | P value | Estimate ± SE | df | t value | P value | Estimate ± SE | df | t value | P value |
| Intercept | −0.944 ± 0.055 | 45.121 | −17.294 | <0.001 | −0.931 ± 0.076 | 27.475 | −12.233 | <0.001 | −0.965 ± 0.043 | 50.309 | −22.497 | <0.001 |
| Tb 25 °C | 0.182 ± 0.019 | 126.803 | 9.323 | <0.001 | 0.399 ± 0.035 | 76.930 | 11.375 | <0.001 | 0.358 ± 0.031 | 113.520 | 11.439 | <0.001 |
| Tb 30 °C | 0.252 ± 0.019 | 126.803 | 12.926 | <0.001 | 0.536 ± 0.035 | 76.930 | 15.276 | <0.001 | 0.581 ± 0.032 | 114.221 | 18.388 | <0.001 |
| Tb 33 °C | 0.371 ± 0.023 | 130.764 | 16.347 | <0.001 | 0.608 ± 0.037 | 77.950 | 16.472 | <0.001 | 0.659 ± 0.034 | 117.084 | 19.490 | <0.001 |
| Tb 36 °C | 0.381 ± 0.024 | 130.492 | 16.061 | <0.001 | 0.709 ± 0.053 | 81.626 | 13.280 | <0.001 | 0.774 ± 0.038 | 121.942 | 20.518 | <0.001 |
| Body mass | 0.760 ± 0.029 | 41.18 | 25.976 | <0.001 | 0.610 ± 0.105 | 23.099 | 5.800 | <0.001 | 0.674 ± 0.030 | 32.223 | 22.657 | <0.001 |

**Note:**
The GLMMs that best explain RMR for the study species. Estimates are presented ± SE.

**Table 3  $Q_{10}$ values for the species across thermal treatments.**

| | $Q_{10}$ (15–25 °C) | $Q_{10}$ (25–30 °C) | $Q_{10}$ (30–33 °C) | $Q_{10}$ (33–36 °C) | Overall $Q_{10}$ (15–36 °C) |
|---|---|---|---|---|---|
| *Crotalus polystictus* | 1.44 | 1.52 | 1.68 | 2.54 | 1.62 |
| *Conopsis lineata* | 2.36 | 1.85 | 1.05 | 3.42 | 2.20 |
| *Thamnophis melanogaster* | 2.3 | 1.82 | 1.16 | 4.09 | 2.11 |

## DISCUSSION

In this work, we analyzed the thermal sensitivity of performance and RMR of three Mexican snake species with distinct lifestyles. Our results support our hypothesis that thermal physiology is influenced by the distinct lifestyles of the species. According to our predictions, we found that the active forager semiaquatic *T. melanogaster* showed the highest levels of performance and mass-adjusted RMR. We expected *T. melanogaster* to be a thermal specialist, but the results show that it can maintain performance over a wide range of temperatures. The fossorial ambush forager *C. lineata* had the lowest levels of performance, as expected, but it also had a relatively wide $B_{80}$, and its mass-adjusted RMR did not differ from that of *C. polystictus*, which is also an ambush forager, but is a more conspicuous terrestrial species. *C. lineata* seems to be well adapted to the thermally challenging environment that it inhabits. *C. polystictus* exhibited the most resistance to high temperatures, having the highest $T_o$ and $B_{80}$.

It is noteworthy that the upper limit of $B_{80}$ of *C. polystictus* and its $T_o$ fall very near its $CT_{max}$ (Fig. 1). This results in a low TSM, considerably lower than those of *C. lineata* and *T. melanogaster* (Table 1). Having a small TSM can be risky, since being active near the $CT_{max}$ increases overheating risk (*Sinclair et al., 2016*). In such a scenario, behavioral thermoregulation strategies such as retreat site selection, can be key adjustments to buffer the effect of high temperatures (*Sunday et al., 2014*; *Muñoz & Losos, 2018*; *Vicenzi et al., 2019*). In this sense, the activity pattern of *C. polystictus* as a sit-and-wait predator is very beneficial, since much of the time it dedicates to foraging activities happens within shelters

and not exposed in the open. This result suggests that *C. polystictus* would need to actively thermoregulate at suboptimal temperatures to avoid overheating, which aligns with the results we obtained in a thermal ecology study made with the same population, as its $T_o$ and $B_{80}$ (36.7 °C) are higher than its $T_{set}$ (23.4–31.4 °C, R. Figueroa-Huitron et al., 2023, unpublished data). This, however, is a pattern that is present in other ectotherms. *Martin & Huey (2008)* developed a model that shows that, since performance drops rapidly at higher temperatures due to the form of TPCs, the optimum body temperature at which ectotherms operate should be at a lower temperature than the $T_o$ that theoretically maximizes fitness. Considering that $B_{80}$ and $T_{set}$ of *C. polystictus* are relatively wide, it has the capacity to avoid overheating risk while maintaining adequate levels of performance at high temperatures.

The $B_{80}$ of the three species (Table 1) is wider than the reported values for several species of snakes and lizards (*Huey & Bennett, 1987*; *Tanaka, 2008*; *Lelièvre et al., 2010*; *Bonino et al., 2015*; *Romero-Báez et al., 2020*), which suggests that they are thermal generalists that can maintain performance over a wide interval of temperatures. Having a wide $B_{80}$ can be beneficial to the performance of *C. lineata* in the cold site of Los Dinamos, since such a characteristic has been found to allow some species to maintain adequate performance levels in suboptimal temperatures (*Gómez-Alés et al., 2018*; *Valdez Ovallez et al., 2022*). For *T. melanogaster*, its wide $B_{80}$ could be a factor contributing to its wide distribution, the widest among our three species, ranging from central to northwestern Mexico. Wide performance breadths have been associated with species with widespread distributions that can inhabit variable environments (*Bonino et al., 2015*; *Dematteis et al., 2022*).

Higher levels of locomotor performance can be linked to foraging mode and type of habit in snakes, with active foragers being distinctively faster than ambush predators (*Whitaker & Shine, 2002*; *Lelièvre et al., 2010*). *T. melanogaster* is an active forager that feeds on aquatic prey in the open and, accordingly, had the fastest swimming speeds in our tests (Fig. 1). On the other hand, *C. polystictus* and *C. lineata* are ambush foraging species (*Meik et al., 2012*; *Raya-García, Alvarado-Díaz & Suazo-Ortuño, 2020*) who require to strike fast but do not require high speeds to chase their prey. Furthermore, active foragers like *T. melanogaster* can be exposed to higher predation risk since they are more active in the open (*Secor, 1995*). Higher locomotion speeds can help counteract this problem and evade potential predators. A positive relationship between locomotor capacity and survival has been reported for *T. sirtalis* and some juvenile lizards (*Jayne & Bennett, 1990*; *Miles, 2004*; *Husak, 2006*). It is important to note that *T. melanogaster* is the only one of our study species that is ecologically adapted to the semiaquatic lifestyle. In this sense, having the highest swimming speed may also be attributed to morphological adaptations. Body shape is an important adaptation for aquatic and semiaquatic snakes (*Brischoux & Shine, 2011*), with a dorso-ventrally elongated body being more efficient to swim. Another adaptation found in the semiaquatic colubrid *Nerodia sipedon* is the ability to laterally compress the posterior half of the body when swimming (*Pattishall & Cundall, 2008*). This allows *N. sipedon* to boost swimming speed without reducing locomotor performance on land, where, like *T. melanogaster*, it thermoregulates and finds cover. Surprisingly, there are no

in-depth studies with *T. melanogaster* focusing on morphological adaptations to the semiaquatic lifestyle (except for eyesight, *Schaeffel & De Queiroz, 1990*), so more research in this regard is necessary to better understand the locomotor capacities of this species.

All three species in this study presented TTB's above 30 °C, which are similar to those reported for other snakes (*Jacobson & Whitford, 1970*, *1971*; *Huang et al., 2007*; *Tanaka, 2008*). In particular, *C. polystictus* had the widest TTB (37.15 °C) and the highest $CT_{max}$ (43.1 °C). This suggests that this species is more resistant to high temperatures. Furthermore, a wide TTB is often linked to species that experience high climatic variability (*Cruz et al., 2005*; *Gutiérrez-Pesquera et al., 2016*). This relation has been studied in the context of species with wide distributions that encompass different environments, and *C. polystictus* can fit into this category. Further studies made at sites with different conditions are desired to corroborate if *C. polystictus* follows this pattern; however, local seasonal variation can give us a hint in this direction. The study site of *C. polystictus*, San Bartolo Morelos, has noticeable thermal seasonality, and the wide TTB of this species can allow it to thrive during winter and the hottest period of the year, in late spring and early summer.

The GLMMs that best represented our RMR data show that it is positively influenced by both temperature and body mass (Fig. 2, Tables 2 and S4). After adjusting for body mass, we found that *T. melanogaster* had the highest proportional values of RMR. This result aligns with our expectations, given the lifestyles of the three species. *T. melanogaster* chases prey actively, a strategy that demands more energy than ambush foraging and results in higher metabolic rates. This is a trend documented extensively in several snake species (*Secor & Nagy, 1994*; *Dupoué, Brischoux & Lourdais, 2017*; *Stuginski et al., 2018*). RMR increased along with temperature; however, the $Q_{10}$ values calculated for each thermal treatment transition show that the proportional increase was not uniform (Table 3). Some studies with squamates have reported a pattern similar to what we found, in which the $Q_{10}$ values are relatively stable at medium temperatures but show a significant increase in metabolic thermal sensitivity at both extreme ends of its temperature range (*Jacobson & Whitford, 1971*; *Davies & Bennett, 1981*; *Watson & Burggren, 2016*). High $Q_{10}$ at lower temperatures could be an adaptation to maintain favorable levels of physiological performance in suboptimal temperatures (*Zaidan, 2003*; *Plasman et al., 2020*) and a mechanism to rapidly transition from an inactive to an active state (*Davies & Bennett, 1981*). On the other extreme of the spectrum, high $Q_{10}$ could facilitate the physiological responses to heat stress. A study with the garter snake *Thamnophis elegans* showed that by increasing their metabolic rates, it ensures the delivery of the required oxygen to their tissues to maintain metabolic and physiological competence when acutely exposed to near-lethal temperatures (*Gangloff et al., 2016*).

The thermal sensitivity of RMR across the whole experiment was more pronounced in *C. lineata* and *T. melanogaster* than in *C. polystictus*, a result not only clear in the $Q_{10}$ values but also in raw data visualized by thermal treatment (Fig. 3). The overall $Q_{10}$ of *C. lineata* and *T. melanogaster* (2.37 and 2.32) are similar to the values reported for *T. elegans* and other snakes (*Finkler & Claussen, 1999*; *McCue & Lillywhite, 2002*; *Zaidan, 2003*; *Holden et al., 2022*). On the other hand, the overall $Q_{10}$ of *C. polystictus*

(1.67) is lower than the mentioned species and the reported values for other species of the *Crotalus* genus (*Beaupre, 1993*; *Beaupre & Duvall, 1998*; *Beaupre & Zaidan, 2001*; *Dorcas, Hopkins & Roe, 2004*). Having a low $Q_{10}$ can provide some advantages for *C. polystictus*, since having a metabolism with low thermal sensitivity can provide a buffer effect to seasonal changes. In the context of climate change, if its metabolism is not greatly affected by rising temperatures, then its maintenance energy requirements may remain relatively stable. Maintaining regular levels of energy requirements is an important factor in buffering the effects of climate change (*Crowell et al., 2021*).

Overall, our results show different physiological adaptations of the three species to the environments they inhabit. Their different activity patterns and foraging habits are closely linked to these adaptations. It is important to note, though, that we studied only one species from each lifestyle. Further studies with more species are necessary to assess the generality of our interpretations. Also, the three species we studied have relatively wide distributions in which climatic conditions can be very variable. Among squamates, there is evidence that thermal acclimation and geographic variation have considerable effects on the thermal physiology of species (*Angilletta, 2001*; *Bruton, Cramp & Franklin, 2012*; *Sun et al., 2014*; *Wu et al., 2018*). Follow-up studies of other populations of these species with different climatic conditions would provide valuable information to complement our study. Continuing investigation with other species would help us better understand the effect of environmental properties on snake physiology in a more holistic way. Finally, we consider that, at a time in which the need for forecasts of the effect of climate change on ectotherms is continually growing, studies like this one can serve as stepping-stones towards generating more robust assessments, complemented with valuable physiological information.

## ACKNOWLEDGEMENTS

We thank Jorge Ayala-Berdon and Kevin Medina-Bello for logistical and technical assistance regarding the use of the respirometry equipment. We thank Antonio Sánchez, Estefanía Trejo, CONANP authorities, and, especially, Moisés Castillo and his family for providing logistical assistance to survey the study sites. Thanks to Edgar Castañeda Huitrón and Aurora Romero Flores for contributing with the elaboration of the wooden lane used to measure performance. For their support on field surveys, we thank Elia F. Díaz-Martínez, Sarai J. González-Ramos, César Verano-Gallegos, J. Isaac Chiu-Valderrama, Brasil Canales-Gordillo, Daniel J. Sánchez-Ochoa, L. Eduardo Bucio-Jiménez, and J. Luis Jaramillo-Alba. This article is a requirement of Ricardo Figueroa-Huitrón for obtaining the degree of 'Doctor en Ciencias', issued by Posgrado en Ciencias Biológicas, UNAM.

### Funding

This research was funded by Dirección General de Personal Académico Universidad Nacional Autónoma de México (UNAM; PAPIIT Projects IA204416, IN217621, IN222418

and IN203516) and Consejo Nacional de Humanidades, Ciencias y Tecnologías (Scholarship number 775220 and Cátedras CONAHCYT project number 883). The funders had no role in study design, data collection and analysis, decision to publish, or preparation of the manuscript.

## Grant Disclosures

The following grant information was disclosed by the authors:
Dirección General de Personal Académico Universidad Nacional Autónoma de México: UNAM; PAPIIT Projects IA204416, IN217621, IN222418, and IN203516.
Consejo Nacional de Humanidades, Ciencias y Tecnologías: 775220 and Cátedras CONAHCYT project number 883.

## Competing Interests

The authors declare that they have no competing interests.

## Author Contributions

- Ricardo Figueroa-Huitrón conceived and designed the experiments, performed the experiments, analyzed the data, prepared figures and/or tables, authored or reviewed drafts of the article, and approved the final draft.
- Anibal Díaz de la Vega-Pérez conceived and designed the experiments, performed the experiments, analyzed the data, authored or reviewed drafts of the article, and approved the final draft.
- Melissa Plasman conceived and designed the experiments, performed the experiments, analyzed the data, authored or reviewed drafts of the article, and approved the final draft.
- Hibraim Adán Pérez-Mendoza conceived and designed the experiments, performed the experiments, analyzed the data, authored or reviewed drafts of the article, and approved the final draft.

## Animal Ethics

The following information was supplied relating to ethical approvals (*i.e.*, approving body and any reference numbers):

Permit CE/FESI/022023/1578 was given by the ethics committee of FES Iztacala, UNAM.

## Field Study Permissions

The following information was supplied relating to field study approvals (*i.e.*, approving body and any reference numbers):

Collecting permits SGPA/DGVS/06935/21 and SPARN/DGVS/00316/23 (Dirección General de Vida Silvestre, DGVS) were given by the Secretary of Environment and Natural Resources (SEMARNAT).

## Data Availability

The data is available in the Supplemental File.

## Supplemental Information

Supplemental information for this article can be found online at http://dx.doi.org/10.7717/peerj.17705#supplemental-information.

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
