# Peer review of "Physiological thermal responses of three Mexican snakes with distinct lifestyles"

_PeerJ, doi:10.7717/peerj.17705_

## Round 0.1 · original submission · Major Revisions

· Academic Editor

Major Revisions

Both reviewers agree that there is merit in your submission. However, they also identified numerous serious concerns, particularly with the statistical approaches and analyses conducted. Based on these reviews, I have recommended Major Revisions. The reviewers have provided numerous comments that must be fully addressed in a revised submission for this to be considered for PeerJ.

Reviewer 1 ·

Basic reporting

The manuscript is interesting and fun to read, and the experiments are well-designed and well-executed. Parts of it need to be rewritten due to changes in the statistics I believe necessary, but I think it is very possible to address most of my concerns, and I’d like to see it published.

Manuscript style, formatting, and overall structure are good. The literature cited in the Introduction is thorough and relevant, but I think maybe you could cite some of the newer literature on habitat/temperature/size effects on RMR/SMR in reptiles, as I feel there has been a surge in recent years. The English is good but contains some small españolisms throughout. I suggest letting a fluent speaker (I’m not a native English speaker) go over everything before the next submission.

The “habitat type” component is present in the title, but absent from most of the manuscript, only mentioned in passing. With 3 species and 3 habitats there isn’t really a way of including it in the analyses, so I suggest either discussing it much more heavily (in the Discussion but also the Introduction), if you think you can flesh it out convincingly – or better just remove it from the title (but still discuss it, as a hypothesis arising from your results). But first, make the statistical changes listed below and see if the habitat differences are still significant…

Experimental design

EXPERIMENTAL DESIGN

The experiment itself is well-designed and well-executed, with sufficient sample size (more than sufficient! this looks like very hard work!) and several good steps taken that manage to avoid some biases common to this type of experiment. I have a few minor doubts, which do not pose a real problem for using these data, but still need to be addressed in-text:

Why are you measuring CO2 and not O2 if the FoxBox measures O2 as well? If the O2 measurements are not too noisy than they are a much better indicator of RMR (do not depend on RQ). If you choose to stick with CO2 I’d prefer to see a sentence explaining this choice.

You report flow rate but not chamber size. This might have a strong influence on the validity of the following two comments:
- If the chamber is small: VCO2 may fluctuate with breathing cycles (especially in burrowers, some species can hold their breath for really long), in which case 100 seconds is a short time to extract an average. I recommend attaching a representative screenshot from Expedata for each species at high and low temperature.
- If the chamber is large: How did you ensure the animal was resting? Just from the stable VCO2? Or were you observing the animal some/all of the time? Small movements of a restless animal might have been smoothed over if the flow is slow and the chamber large.
* * *
STATISTICAL ANALYSIS

While the general questions tested, and the overall statistical approach is sound, many parts of the statistical analysis of RMR are flawed. I think that a complete re-analysis of the same data with a number of important changes (listed below) will not weaken the conclusions but rather strengthen them. Anyhow these changes are crucial in some places where the structure of the statistical model is not the best or even incorrect. Some further analyses can also be attempted with this data without much effort which will enrich the interest of the findings (see comment on Q10 calculation).

Both RMR and body mass should be log-transformed in the RMR model because their relationship is usually an allometric power relationship. I think this is the major cause of problems further down in the results. Using a linear RMR is problematic also for the relationship with temperature in this same model (remember what the Q10 equation looks like).

Treating the relationship as linear causes yet another error when you compare species and try to “correct” for mass by dividing, which is misleading if the relationship is a power (which I suspect it is). The smallest species might have the highest RMR/mass simply because the allometric slope is smaller than 1. This would be the result in almost any set of species with such notable size differences. I suggest that for this comparison put all 3 species in one model with species as a fixed factor, along the lines of: log10(RMR)~log10(mass)+species, OR: log10(RMR)~log10(mass)*species if you see the allometric slopes are very different. This will give you the real difference between species. If there still is a difference – I suspect there will be – now you can say that it is because of habitat differences.

Why is temperature a discrete and not a continuous variable?? If you later state that RMR rises continuously with temperature (and testing Q10 additionally implies this) than why not report the slope with which it does so? I think that since the effect is non-linear, log-transforming as suggested above paired with temperature as a continuous numeric value will make this model both easier to interpret and objectively a better model.

Why is Q10 calculated only between the highest and lowest? It would be very interesting to see if Q10 is the same between all temperatures or if it shifts at some point (there’s an Arrhenius breakpoint at a certain temperature for many biological processes. Worth checking).

How is skewness calculated? from what I understand you compare AIC (and R2 - do these two always agree?) between various models which each have a different distribution shape. Why not, whether instead or in addition, calculate the skewness directly? You can use the skewness() function in the R package ‘moments’ and obtain an objective numeric value (what’s known as the 3rd moment) for how skewed the data are. If using a set of curves is something commonly done, than please cite others who did it similarly. If not, I’d just report the skewness value.

Validity of the findings

The findings are valid overall. The one serious misgiving I have is that there are obvious mistakes in the raw data for swim speed. It doesn’t look like data forgery to me, definitely not, more like something that got messed up while working in Excel. However, the paper’s publicability (at least the swim speed section) depends on the authors being able to procure their original data. There are many occurrences of perfectly replicated values at different temperatures or individuals, in all three species. Mostly these are in adjacent cells but sometimes very far apart (e.g. C. polysticus ID 21 and ID 33 at 36°C). This looks like an honest mistake to me, but must be solved for these data to be used in a publication. Go back to your original data and try to find out when and where the error occurred, and fix this. I also admit a slight curiosity regarding the high abundance of round numbers in the swim speed data… how were the measurements done to achieve both a round 30 cm/s and 26.0869565217391 cm/s, for the same individual?

Specific results are reported in a satisfactorily, but when describing the overall findings of the study, these must be rephrased in several places. For example (but this is repeated several times including the Abstract) in lines 346-352. The temperature dependence of RMR itself is hardly a result, it almost goes without saying in ectotherms, since it’s rooted all the way in the animals’ biochemistry.
Same for mass dependence – if an organism is bigger, then it has more cells, then it has more cells metabolizing, then it uses up more energy. That’s why sentences like “RMR of juveniles was lower than that of both males and females, a discrepancy that is related to the considerable size difference between adults and juveniles” read a bit oddly. It’s not a discrepancy, it makes as much sense as “big coconuts weigh more than small coconuts”. Is their specific RMR (RMR per mass) higher or lower? Read a bit more about allometry, like things by Glazier, White, Nagy and so on and incorporate it in your conclusion-making.
Your results are therefore not “this relationship exists” but rather, how this relationship manifests in these species. The manuscript mostly treats this accordingly when reporting each result in itself, but rephrase all the general statements to reflect what is the actual discovery in the study, and see previous comments on the statistical analysis.

Maybe I missed this, but I don’t understand how you arrived at the conclusion that sex-dependent differences are only due to unsexed juveniles (line 356). In the tables it looked like there was an effect. I must have missed something so please make it clear so others don’t miss it as well.

I wholly support the choice to use swimming rather than sprinting speed. However, when discussing why T. melanogaster is the fastest swimmer you ignore the fact that it is more aquatic than the other species (the others rarely meet water in their usual routine). It could be a biomechanical adaptation for better swimming which is unrelated to hunting style and predation risk.

The underlying RMR data are provided and seem to be of high quality, so will probably support the conclusion if the authors improve the statistical analyses.

Additional comments

Some of the figures that show trends from the models, particularly Fig. 3, would benefit much from also presenting the scatter of the data points.

Reviewer 2 ·

Basic reporting

The most important question I have after reading the introduction is, how novel is this work? TPCs, locomotor/RMR response to temperature has been extensively studied for nearly a century. The knowledge gap that this study addresses needs to be more obvious. The authors provide a thorough explanation of thermal tolerance, but little if any explanation of what they mean by thermal sensitivity. The introduction could be improved by explicitly describing the difference between the two. Similarly, I think the justification for assessing RMR is well-defined, but the justification for locomotor performance is less so. The authors should make a clear case as to why locomotion is being assessed in this study.

Specific recommendations listed by line below.

a. Clear, unambiguous, professional English language used throughout.
i. Line 50 The authors should state whether birds are included in their assessment of reptiles.
ii. The authors do not use consistent language throughout text. On line 55, "tolerance breadth" is used, but then shortly after, "tolerance interval," then on line 61 "thermal tolerance." Throughout the report the terminology changes but should instead keep the same terminology consistently throughout the report.
iii. Line 176 Use consistent language. Here the authors use "critical thermal range" but elsewhere they use "thermal tolerance breadth"
iv. Line 64 Be more specific than "not informative enough," which, ironically, is not informative enough. CT max/min is not informative enough relative to?
v. Authors should change “analyzes” to analyses at all occurrences. I noticed this typo on lines 71, 237, and 256 but the authors should double check it did not occur elsewhere.
vi. Line 94-97 is a run-on sentence.
vii. Line 294: If I am understanding the authors' meaning correctly, "with" is a more appropriate word to use here

b. Intro & background to show context.
i. Line 77: Exponential increase has a specific curve which does not apply to the relationship between RMR and body temperature. "Linear increase" would be the proper description.
ii. The authors do not provide a satisfactory rationale for the selected species. They represent different environmental niches, but so do many other snakes. It should be made more clear why these specific species were chosen over another species.

c. Literature well referenced & relevant.
i. Line 73 I think the authors have misinterpreted this article, titled: "Cautionary notes on the descriptive analysis of performance curves in reptiles." The authors specifically advise against certain methodologies used in the present study. I would suggest authors remove the reference here since it does not support the statement. Then in the statistical methods they should briefly address some of the common pitfalls in this type of study as discussed in the article and how their study avoids these issues.

d. Structure conforms to PeerJ standards, discipline norm, or improved for clarity.
i. No comment

e. Figures are relevant, high quality, well labelled & described.
i. The figures chosen are good representatives of the results. Personally, I did not find the tables to be useful but I am sure there are other readers that would appreciate them.
ii. Personal preference, but I think that Figure 1A-C could be improved by either changing the vertical alignment to a horizontal alignment, or ideally, changing all three figures to a single figure. This will allow easier comparisons among species. I think if proper care is taken to ensure the figure is not too cluttered, a single figure would be best. If not possible, I think a horizontal alignment would help to reduce the repetitive y-axis label. If kept as three figures but changed to horizontal alignment, that would mean that x-axis (body temperature) becomes repetitive, but I think that would look much “cleaner” overall.

f. Raw data supplied (see PeerJ policy).
i. Authors stated data will be made available upon submission however I did not see an access/reference identification.

Experimental design

a. Original primary research within Scope of the journal.
i. See previous comment regarding originality of work.

b. Research question well defined, relevant & meaningful.

I found it very difficult to determine what the purpose was for the paper. The authors seemed to state multiple similar aims and I’m not sure if they are meant to be interpreted as being the same or distinct topics. Predicting the effects of temperature on reptile ecology, both currently and with anticipated climate change, is mentioned but then they never circle back to explain how their results contribute to understanding this topic. Later in the introduction, the authors state they are interested in assessing how reptiles are adapted to their environment and ways in which temperature influences their physiology. The concluding statement of the intro states the objective as using multiple species with distinct behavioral/ecological characteristics to better understand how temperature affects physiology.

i. The introduction concludes with the study objective, which is "[. . .] to assess how reptiles are adapted to their environments and how they could respond to ongoing and future climate alterations." The authors assess the suitability of the reptiles to their habitats, but do not place their results in the context of ongoing/future changes associated with climate change. To improve, the authors could supply annual temperature range for all three snake species and perhaps anticipated changes related to global climate change. Then place their results in the context of current vs. future effects, based on thermal biology.
ii. From my interpretation of this manuscript, I believe the study purpose is more accurately stated as assessing the suitability of animals to their environment according to their thermal biology. However, there are limited inferences that can be made since laboratory results often do not reflect behaviors in the wild, and the experimental temperatures are outside the range of what the animals' typically experience in their natural habitat due to behavioral thermoregulation.
iii. Lines 89-91: I agree with the statement of importance

c. It is stated how the research fills an identified knowledge gap.
I did not find a clear explanation of how this work is novel and/or what knowledge gap it addresses.
i. Line 64: Investigations of TPCs are not a "recent" interest. These types of studies have been ongoing for close to a century.

d. Rigorous investigation performed to a high technical & ethical standard.
i. The authors attempt to use experimental temperatures that snakes experience in their natural habitat, however, it appears they have used ambient environmental temperature data, rather than body temperature of the animals themselves, which can be very different from the environment in behaviorally-thermoregulating ectotherms.
ii. The methods do not describe critically important aspects of snake husbandry. For example, it is not stated what temperature were the snakes kept at, whether they were provided with an appropriate light/dark schedule, whether they were given access to shelter or enrichment within their enclosures.
iii. Feeding regimen was not described (e.g., What was each snake fed, what percentage of body mass was each meal, how frequently were they fed before release?).
iv. Did the authors perform any measure to assess whether the condition of the animals had deteriorated during captivity? For example, the mass on day of capture vs. day of release? There should be some metric of assurance that the animals remained in good health throughout the experiment.
v. While CTmin/max and the righting response is widely used, I would advise the authors to consider using voluntary thermal max/min (generally, the temperatures at which the animal begins looking to escape the temperature treatment; see Rozen-Rechels et al. 2019) in future studies of a similar nature. This is considered by some as a more ethical and informative measure as it takes into account the animal's perspective.
vi. Line 160 the authors state only some animals were used for the CT min/max experiments but do not explain why. They should move the list of sample sizes here from line 163-164.
vii. Line 162 missing specific epithet.
viii. Line 167 Were these metal forceps? This sounds like an uncomfortable method of handling snakes, especially the larger ones.
ix. Line 176 In the future, a more gradual cooling methodology would be desirable
x. Line 222: Since the snakes were kept in darkness for the respirometry tests, I wonder if the authors were able to observe the snakes and ensure they kept relatively still for the duration of the trial.
xi. Lines 226-227: Specify the quantity of snakes for each species that represents 2023 captures.
xii. Line 240: Specify what "population mean" represents, i.e., how are populations differentiated? By species, by sex, by age, etc.
xiii. Line 242: Is the CTmin/max from the earlier experiment or individualized per snake? Were differences based on sex/age/time accounted for?
xiv. Lines 250-251: Mass and SVL are not independent variables. I would recommend using BCI (residuals of mass vs. SVL) or choosing one over the other.

e. Methods described with sufficient detail & information to replicate.
i. Lines 140-142: More details needed on how juveniles were assessed as too small for cloacal probing, and how it was determined whether a female was pregnant.
ii. For the CTmin/max experiment, how much time did snakes get between hot/cold trials? Was there any variation in the order of treatment?
iii. Line 173 I have never seen a wild nor captive snake pant, perhaps this should have been the CTmax? How close did the onset of panting occur to the loss of righting response? See comment regarding voluntary thermal max/min.
iv. Lines 181-182: The authors should be more transparent by explaining what exactly they mean by "previously obtained thermal ecology data." Is this data from the present study or some other time? How were the values determined, particularly for C. lineata, whose natural habitat annual temperature range is 9-15 degrees Celsius (lines 117-119)?
v. Lines 182-185: It’s unclear how the authors obtained this data and whether the data is trustworthy.
vi. Lines 186-188, Line 661: There was no raw data for 36 degree Celsius temperature trials. Perhaps the authors decided against using this data after all? The reasoning is somewhat questionable - if the aim is to study how climate change might affect these snakes, testing an arbitrary value higher than body temperatures voluntarily maintained by the snakes seems superfluous.
vii. Line 196: While I understand the authors train of thought, I don't think it is accurate to state stress was reduced by extending the duration of the experiment. Keeping wild animals for extended periods can also increase stress, in particular chronic stress from extended captivity vs. acute stress from handling during an experimental trial.

Validity of the findings

a. Impact and novelty not assessed.
i. No comment (I am not sure what is meant by this criterion).

b. Meaningful replication encouraged where rationale & benefit to literature is clearly stated.
i. Line 333: Objectively untrue. Ambush predators by definition must be extremely quick-moving to successfully obtain prey. However, they do not require substantial endurance.
ii. Lines 333-335: Is this supported by the literature?

c. All underlying data have been provided; they are robust, statistically sound, & controlled.
i. Lines 261-264: This is misleading since the first result is thermal tolerance breadth, which did not include all of the snakes listed but rather a subset of each species collected in July 2023.
ii. Line 264: TTB should be a range. I don't know what the single temperature is meant to represent (mean? mode? optimal? etc). The authors should state what the single temperature value is a measure of.
iii. Line 274: Is this every study animal? If so, there's no need to restate sample sizes if they are already in the methods.
iv. Lines 282-285: Please include the metabolic rates
v. Lines 296-297: While this may be true as a generalization, we do not know, and the authors do not provide evidence, that this is true for their experimental animals. I would suggest either removing this portion entirely, moving it to the discussion, or justify it by including average mass per species.
vi. Lines 329-331: Was this at every temperature? Add a reference to the appropriate table/figure.

d. Conclusions are well stated, linked to original research question & limited to supporting results.
i. Lines 299-302: I don't understand why the authors begin the discussion with this statement. It does not pertain to their study objective and was not mentioned in the intro nor their methodology. These topics are not mentioned anywhere else in the text.
ii. Line 304: How are we to know this is reliable data if it is unpublished, and the current study did not measure this value? Further, I am not sure what the authors mean by "set point of preferred temperature."
iii. Lines 306-308: Cite the corresponding figure and/or table
iv. Lines 307-308: Another idea that has not been addressed anywhere else in the manuscript. The authors should at least mention this idea in the introduction since TSM could become very important in light of climate change, presenting a possible disadvantage for future populations of C. polystictus
v. Line 309: his? Likely a typo
vi. Line 313: Site selection is part of behavioral thermoregulation strategy.
vii. Lines 317-318: Poor interpretation of data that is not justified by the present study or relevant literature. The optimal and preferred temperatures obtained in a laboratory cannot attest to how well adapted an animal is to their environment. See Angilletta, M. J., & Werner, Y. L. (1998). Australian geckos do not display diel variation in thermoregulatory behavior.
viii. Lines 318-319: Since B80 represents a range of temperatures, why is only one temperature value included?
ix. Lines 323-329: Needs more support for these claims from the literature
x. Line 340: Authors should explain what is noteworthy about their result.
xi. Line 340. As stated by the authors, thermal tolerance is a RANGE of temperatures, so again I am puzzled as to why only one temperature is reported.
xii. Line 341: Aren't T. melanogaster and C. lineata the more widespread species in this study?
xiii. Lines 343-345: Is the point the authors want to make that despite being less widespread, as is typical for species with a wide thermal tolerance breadth, a highly variable climate may be the driving selector for wide TTB? Regardless, the authors should make their point more explicitly within the text.
xiv. Lines 362-363: The authors describe thermal sensitivity being similar in “[. . .] other snakes and a general sample of squamates [. . .]” This is redundant as all snakes are squamates.
xv. Lines 369-371: I do not understand what the authors mean by this sentence.
xvi. Lines 373-376: The authors state that their result of C. lineata having the highest mass-specific RMR is in contrast to prior research since C. lineata is a fossorial species whose energy requirements would be similar to ambush hunters. However, the articles cited (specifically Dupuoe et al. 2017) classify only boids and vipers as ambush foragers, i.e., C. lineata would be considered an active forager. So their result actually fits with the cited studies rather than contradicts them.

Additional comments

The authors found that snakes inhabiting diverse ecological niches appear to have temperature-related characteristics that are well-suited and possibly adaptive to their respective habitats. Overall, I think there are interesting findings in this manuscript and they are worth publishing. However, the manuscript needs substantial work.

One major flaw with this manuscript that was noticeable throughout was the inconsistent use of specific terminology and abbreviations. I think the authors should thoroughly proofread to ensure: 1. consistency, 2. that the abbreviations are stated at the first use of the word or phrase, 3. and are not repeated later in the manuscript. An example was using the terms tolerance breadth/tolerance interval/thermal tolerance/TTB interchangeably without cluing the reader in that these are all supposed to be the same thing. Another example would be the use of performance breadth/B80/ in figure captions, where it is defined in some captions but not others. Swimming speed as a dependent variable is introduced on line 198. Vmax is meant to represent swimming speed, however the abbreviation is not listed until line 246. A list of abbreviations at the beginning of the manuscript might also be helpful.

I think the authors should develop a comprehensive hypothesis along with predictions for each species, based on known relationships between ectotherms and the environment (resting metabolic rate vs. standard metabolic rate, behavioral thermoregulation, cold metabolism, foraging strategies energy use, etc.). It would provide a clear direction for the introduction and would better prepare the reader for the information presented later in the text.

My concern with the statistical modeling is in the linear models that used both mass and SVL as fixed effects. These are not independent variables. The methods are straightforward and easy to understand, as are the results. The figures chosen are good representatives of the results. Personally, I did not find the tables to be useful but I am sure there are other readers that would appreciate them. All references on the reference list have corresponding in-text citations.

The discussion has similar issues of maintaining focus as did the introduction. The first sentences of the discussion bring up topics (thermal coadaptation, preferred temperature which was not tested and is not the same as optimal temperature, which was tested, and thermal safety margins) that are not mentioned prior to the discussion. Perhaps the authors could introduce the thermal coadaptation concept in the intro and make their hypothesis/predictions in this context.

The discussion lacks a clear integration of the authors’ results into existing scientific knowledge. The authors claim their work is similar to/in contrast to other studies and cites those studies, but in at least one case (described above) they had incorrectly identified their result as being contradictory to the cited cases.

---

## Round 0.2 · Minor Revisions

· Academic Editor

Minor Revisions

Thank you for this much-improved version of your manuscript. Both reviewers agree that this version is much stronger than the original submission and were generally satisfied with your responses to their comments. Nonetheless, the reviewers have now made a handful of additional, mostly minor, points that need to be addressed before this manuscript can be taken further. Most of the points are to further clarify the language used and more clearly describe your approach to this study; thus, they should be easily dealt with overall. I look forward to seeing a revised version fo your manuscript.

Reviewer 1 ·

Basic reporting

The article is very good in its current form, a commendable improvement from the first version. It is now coherent and to-the-point, and the introduction and discussion are well related to the experiment and its results.

Experimental design

The experiment itself was well-designed, as I wrote in the first round, and I was impressed by the protocol for including many precautions to avoid common pitfalls.

But allow me one small rant, regarding the TPCs. I understand AICc is a method used by others to define the curve, but for the purpose of what you are doing, I insist you calculate the skewness and not just tell me what type it is from the pool of models. The problems with your AIC method are a) it is not quantitative, b) it does not provide one definite curve, c) you only perform it on one individual, so most of your data is wasted.
Let’s look at the T. melanogaster in Supplemental Table 2, which you later go on to claim has a “bell-shaped” curve (e.g. line 344). I agree with you on that, but not because of the model. First, AIC theory is that all models within delta-AIC of <2 explain the data equally well, and here, all three top curves are within delta-AIC of 0.596. So is it skewed and not skewed at the same time? Your method makes it more confusing than it needs to be.
Anyway, if I take any of them: “Extreme Value 4-P Fronted”, “Exponentially modified Gaussian”, “Log Normal-4P” - these don’t mean anything that I (nor *you*) can understand about how the performance is distributed. They are shape families but not the shape itself. Exponentially modified? OK, well, with what exponent? Is the modification of any ecological significance or just a tiny mathematical quirk? AIC doesn’t tell you (the plot helps here, but you need something more objective).
And lastly, if I understand correctly, you tested the curve AIC just for the individuals with highest R2?? What if everyone else had a different distribution? You have such a beautiful sample size and you don’t use it here and that’s a pity.
I suggest a very simple alternative method: just calculate the TPC skewness for each individual. Then report the average skewness for each species. Done. The plot can stay the same (but would be better if you can superimpose all individuals together!).

Validity of the findings

The findings are valid, and their interpretation and discussion are well supported and require only minor tweaks (see general comments).
I am happy that the authors are confident in their raw data and have given a satisfactory explanation for why they appeared odd to me. Supplemental Table 1 is really really good and helps understand the procedure.
However, I still think the incidence of *repeated* values is higher than expected from the 60fps frame rate, especially when they are often subsequent, and I encourage you to double-check at least a subsample of these values and make absolutely sure you didn’t accidentally analyze the same video twice. I will be 100% convinced if I see something exactly like what you did in Supplemental Table 1, but for C. polystictus #9 in the treatments 25°C and 30°C… or #25 at 30°C and 33°C…

Additional comments

Title and line 139: are the lifestyles truly “contrasting”? Maybe “disparate” or “different” would be more precise.
Line 23: not sure “across” is the right word here.
25, 61: you are not discussing susceptibility to temperature in this context, you are discussing sensitivity.
34: general*IZED* linear mixed models.
39: aligns WITH.
45: why are you talking about “every subsequent treatment” rather than an increase with warmer temperatures? I respect your choice of treating treatments as discrete for the sake of statistics, but for the sake of summarizing the results or discussing the underlying biological phenomenon - this is simply temperature dependence and you can say so (what more, the treatments were not subsequent due to the randomization).
45, 47: not sure “augmented” is the right word here. Maybe “increased” would be more precise.
50: it is not clear what you mean by “different levels of physiological adaptations”.
65-66: this can be explained more clearly, as I wasn’t sure whether this is a definition or an example of thermal sensitivity. Perhaps: “Highly thermally sensitive species are those whose physiological traits covary strongly with temperature in contrast to species with lower thermal sensitivity …”
71: put the word reptiles in quotation “reptiles”.
73-74: the phrasing “metabolism, primarily evaluated by the metabolic rates of O2 consumption or CO2 production” is confusing. Maybe “metabolic rates, primarily evaluated by the rates of O2 consumption or CO2 production”.
96: unclear if by “Thermal variation can affect metabolic rates at different rates” you mean among different species or within species at different conditions (you later give examples of both, but make it clear in this sentence already).
99: change the phrasing “increase activity patterns” to “maintain high activity levels”.
101: what does “respond” refer to? Make it clear when you are talking about metabolic rate.
104: Dabruzzi, Sutton & Bennett, 2012 is not really an example of plasticity or acclimation as you write, high Q10 simply allows the snakes use less energy when submerging. Do cite this reference, but change the phrase to reflect that it is an example of a highly thermally sensitive species and how it helps it cope. I have some examples of acclimation and plasticity in snake Q10, if you are interested, but I don’t think it is really the subject of the paper.
107: “A Q10 value of two, for example, represents that the metabolic rate has doubled” – this explanation is incomplete. Q10 specifically measures what is the change with an increase of 10 degrees Celsius (and the calculation in line 303 corrects for it if the gap is different).
114: I am unsure if you are really discussing “microhabitats” and not just “habitats”, as some of your later examples deal with macroclimate rather than immediate conditions.
123: What about fossorial and aquatic species? This would be a good place to mention them. There might be some relevant literature and it will support your prediction and results. Currently there is not enough introduction leading up to the results about these two modes (see line 144: prediction makes sense but should be supported by literature, e.g. Andrews & Pough 1985 in Biology of the Reptilia). Giacometti & Tattersall (2023 in Evolutionary Ecology) tested if fossorial amphibians had lower RMR (they didn’t), and Luna et al. (2017 in Comp. Biochem. Physiol. A) tested if fossorial rodents had lower BMR (they did). I can’t think of a reference for aquatic vs terrestrial species, but please search the literature for it, and mention them in this paragraph.
128: write “in Mexico” not “this country”.
183-184: excluding winter was good practice even if you could have found the snakes, since most snakes’ metabolic rates decrease in winter beyond the effect of temperature due to hibernation, so would have biased the results. Maybe mention this too.
248: registered by you in the field? Or registered ever in the field? Unclear.
249: where is the data on Tset from? Provide a reference, or state if it is your data.
256: size of the species or of the individual?
258: good. randomization is very important.
261: worth mentioning that RMR during activity hours is higher than SMR during resting hours (Andrews & Pough 1985 in Biology of the Reptilia, Dubiner et al. 2023 in Animal Ecology), therefore, it is good practice that you didn’t mix the two rates.
282: “determined the fps that the snakes took to move” is incorrect. You determined the frame count, fps was constant (60).
299: did you correct CO2 with a reference chamber? If not, are you sure CO2 in the entering air was negligible? Can you provide the equation (or its number) in Lighton 2008 you used for the calculations?
311: small samples? What were the sample sizes? You report them in the results but I would like to see mentions of sample sizes in the methods everywhere that it is relevant.
328: if you used AICcmodavg, did you do any model averaging? Because this was not mentioned. Generally, you should mention these packages in the relevant places or mention here what you used each one for.
341, 344: nothing in your methods allows you to claim “left-skew” or “bell-shape”, whereas your actual curve results don’t appear here (naturally, as they are impossible to understand). See my long comment on TPCs and skewness.
341, 345, 361-362: write in parentheses what their B80, To, and Q10 values were.
375: also Muñoz & Losos 2018 in Am. Nat.
378: note that species living at warmer body temperature have more left-skewed body temperature (Huey & Pianka 2018 in Functional Ecology, Dubiner et al. 2024 in GEB) which aligns with the skewness of TPC. Matching the TPC and temperature skewness leads, if integrated over time, to a greater portion of activity does coincide with the thermal optimum for activity. This is noted in the Martin & Huey paper you cited and is important to show that the skewed TPC is not a maladaptive trait.
394: there are shortfalls to the discussion, especially that you only have one species representing each lifestyle, so linking the differences to species traits should be done with caution, and you should state this explicitly and clearly, I suggest at the end of this paragraph in 2-3 sentences.
407-410: I’m not sure the ventral scales are important to this discussion, especially as C. lineata has more scales… what about the overall body shape? Swimming is easier with a more ribbon-shaped body whereas rattlesnakes are flat and fat, and fossorial species are cylindrical, right? But before any mechanistic explanation, I would extend the sentence 405-406 saying that it is expected to be a better swimmer simply because of its ecology, being the only one that actually needs to be able to swim and have morphological and behavioral adaptations to do it.
411: snakes don’t have a “lower half”. Posterior? Ventral?
429: not “correcting by”, maybe “adjusting for”?
435: just say “the results show…” etc.
436: see comment for line 45 about discussing “subsequent treatments”.
470: extend “continuing investigation with other species” to a separate full sentence as this is very important for testing the generality of your results.

Reviewer 2 ·

Basic reporting

i. There are some minor grammatical errors.
ii. Introduction/background information is relevant and important for the study. If possible, it would be better to place predictions in context of important terms and abbreviations that were mentioned in the intro and that are actually measured in the study (e.g., highest/lowest RMR, TSM, Q10, etc).

Experimental design

a. Original primary research within Scope of the journal.
i. The revisions made by the authors have made the originality of their research clearly evident (lines 127-133).
b. Research question well defined, relevant & meaningful.
i. Objective is stated on lines 129-133. The relevance and meaningfulness of this research to the broader scientific community should be more obvious. As of right now, the objective is very “snake-centered” and it would be better if the authors could indicate how their research is valuable beyond just snakes.
c. It is stated how the research fills an identified knowledge gap.
i. Yes; lines 127-129.
d. Rigorous investigation performed to a high technical & ethical standard.
e. Methods described with sufficient detail & information to replicate.
i. The revised Methods have been written in much greater detail.

Validity of the findings

i. In its current state, the beginning of the discussion reads very abrupt. The discussion should begin by restating the purpose of the experiment and the authors’ hypothesis/predictions and whether or not they were supported. It would also be helpful to briefly restate the defining characteristics of the three species. Since the results are ultimately compared to the ecology of each species, it’s important to maintain that as the overall focus of the discussion.

Additional comments

a. The authors have made significant improvements in making the purpose of their study clear and in writing descriptive, detailed methodology.
b. The figures have improved immensely. They are aesthetically pleasing and extremely informative.
c. The abstract should have only a limited recap of results. Beginning with the sentence “This suggests [. . .]” on line 36 and continuing through line 49, the results need to be summarized more succinctly. Otherwise, the abstract reads well.
d. The conclusion does not have the same structure and organization as the earlier sections. The authors should revise the conclusion so that the overall purpose of the study remains as the focus. The discussion should progress logically from the first main result to next main result, to next main result, etc.
e. The authors should also avoid oversimplifications (e.g., lines 372-373, “Animals with smaller TSMs have more chances to overheat, and possibly die, as the climate warms (Sinclair et al., 2016).” The animals have just as many chances to overheat in the current climate. However, the difference and potential danger with future climate change is that the animals may have no way to cool themselves down, if their environment cannot offer areas of sufficiently low temperatures.
f. It would be helpful to include a short summary paragraph at the end of the discussion to re-emphasize the most important findings.
g. The article would greatly benefit from a concluding statement that encompasses a wider audience than just those interested in snake physiology.

---

## Round 0.3 · accepted · Accept

· Academic Editor

Accept

Thank you for your efforts to revise this manuscript during the 2nd round of reviews. I feel your revisions have made for a much-improved submission and, thus, I think that your manuscript is ready for publication in PeerJ.